# Bias and negative values of COVID-19 vaccine effectiveness estimates from a test-negative design without controlling for prior SARS-CoV-2 infection

Ryan E. Wiegand [1] ✉, Bruce Fireman[2], Morgan Najdowski[1], Mark W. Tenforde [3], Ruth Link-Gelles [1] & Jill M. Ferdinands[3]

Test-negative designs (TNDs) are used to assess vaccine effectiveness (VE). Protection from infection-induced immunity may confound the association between case and vaccination status, but collecting reliable infection history can be challenging. If vaccinated individuals have less infection-induced protection than unvaccinated individuals, failure to account for infection history could underestimate VE, though the bias is not well understood. We simulated individual-level SARS-CoV-2 infection and COVID-19 vaccination histories and a TND. VE against symptomatic infection and VE against severe disease estimates unadjusted for infection history underestimated VE compared to estimates adjusted for infection history, and unadjusted estimates were more likely to be below 0%, which could lead to an incorrect interpretation that COVID-19 vaccines are harmful. TNDs assessing VE immediately following vaccine rollout introduced the largest bias and potential for negative VE against symptomatic infection. Despite the potential for bias, VE estimates from TNDs without prior infection information are useful because underestimation is rarely more than 8 percentage points.

Test-negative designs (TNDs) are an indispensable tool for assessing vaccine effectiveness (VE). TNDs were designed to assess VE against symptomatic infection of seasonal influenza[1,2], but have been used to estimate VE against SARS-CoV-2 symptomatic infection[3], emergency department or urgent care encounters[4], hospitalizations[5], invasive mechanical ventilation[6], and death[7] and to support policy decisions[8]. A TND can be performed rapidly, at lower cost than other studies, and with reduced confounding from health care seeking behavior compared to other observational study designs[1,2]. The efficiency and feasibility of a TND comes with many challenges[9,10], especially regarding the assumptions of how cases and controls are ascertained; controls should be representative of the source population that yielded the cases[11].

Protection from infection-induced immunity can present challenges when estimating VE from a TND. Participants' history of prior SARS-CoV-2 infection has often not been incorporated into VE studies[11]. For COVID-19 studies, infection history data is not collected due to self-testing, asymptomatic infection, and mild infections not requiring medical attention[12]. Bias can arise if prior infection status is misclassified[13] or not accounted for in models[14] and could result in a VE estimate below zero[15]. Serologic testing has been recommended to correct this bias[14] but possesses many challenges, including decreasing sensitivity due to antibody decay[16], potential inability to detect past infection in people with a current infection, increased cost, and decreased power[17] since over 87% of the US population had detectable

[1]Coronavirus and Other Respiratory Viruses Division, Centers for Disease Control and Prevention, Atlanta, GA, USA. [2]Kaiser Permanente Vaccine Study Center, Kaiser Permanente Northern California Division of Research, Oakland, CA, USA. [3]Influenza Division, Centers for Disease Control and Prevention, Atlanta, GA, USA. ✉e-mail: rwiegand@cdc.gov

SARS-CoV-2 antibodies from infection in October–December of 2023[18]. Additionally, many people have had multiple prior SARS-CoV-2 infections, and serologic testing does not provide information on the number of total infections nor the time since or variant of the last infection, which are important for understanding the potential impact of past infection on VE.

Considering these challenges, we endeavored to assess the bias in VE against symptomatic SARS-CoV-2 infection and severe disease from a TND when prior infection is unaccounted for in analyses. Micro-simulations were created based on the COVID-19 pandemic, where each person's vaccination and infection history was generated up to May 2023, followed by a hypothetical vaccination campaign and TND to estimate VE against symptomatic infection or severe disease. Multiple parameters relating to vaccine and infection protection waning and the TND study design were varied.

## Results

Results from all simulated parameter sets and aggregated estimates are in the Supplementary Excel File.

### VE against symptomatic infection

Per simulated population, the median protection against symptomatic infection at the end of the historical period ranged from 0.26 to 0.51 (where zero was no protection and one was complete protection) and, on aggregate, the distribution of median protection against symptomatic infection was lower when infection protection completely waned by 72 weeks compared to 96 weeks (Supplementary Table 1). The distribution of median protection against symptomatic infection by similar across the number of vaccinations (Supplementary Table 2) but increased with increasing number of infections (Supplementary Table 3).

VE against symptomatic infection (Fig. 1, panels a, c, e) in unadjusted models was highest for people 1–2 months since vaccination

(VE = 46.3%; CI: 45.6, 47.0) and decreased with more months since vaccination, reaching the lowest at 5–11 and 12 or more months (VE = −1.6%; CI: −1.9, −1.3). VE against symptomatic infection was also lower the more months included in the recent vaccination exposure, the longer time since vaccination, and the fewer number of total vaccination doses. Distributions of estimated VE against symptomatic infection tended to be wide and cover a wide range of VE values, except for exposures with VE against symptomatic infection estimates close to zero which had narrow, unimodal distributions (Supplementary Fig. 1).

For each exposure definition, at least 94.6% of simulations possessed a bias of VE against symptomatic infection in unadjusted analyses of 8 percentage points (pp) or less (Fig. 2, panels a, c, e). All vaccination definitions excluding 5 vaccination doses and 3–4 months since vaccination had at least 95.4% of simulations with a bias of VE against symptomatic infection less than 6 pp. Sensitivity analyses utilizing only simulations when the TND happened during or after the vaccination rollout possessed similar results compared to the results from all simulations (Supplementary Fig. 2). Mean bias was at most 5.5 pp for any exposure definition (Supplementary Fig. 3).

For the exposure of vaccination in the previous 3 months (Fig. 3), the overall mean bias was −1.4 pp (CI: −1.5,−1.3). Bias was higher when hybrid protection was defined as the greater source of protection boosted by 30% (Bias = −1.7 pp; CI: −1.8,−1.6) and lower when the greater of VP or boosted by 30% of IP or IP boosted by 10% of VP (Bias = −1.1 pp; CI: −1.2, −1.0).

The timing of the vaccination rollout and TND also impacted bias. For people vaccinated in the previous 3 months, the largest bias occurred when the vaccination rollout happened immediately before the TND (vaccination rollout in weeks 1–12 and TND in weeks 11–22: Bias = −1.9 pp; CI: −2.1, −1.7; vaccination rollout in weeks 11–22 and TND in weeks 21–32: Bias = −2.2 pp; CI: −2.4, −2.0) and the smallest bias was when the vaccination rollout and TND took place in weeks 1–12 and

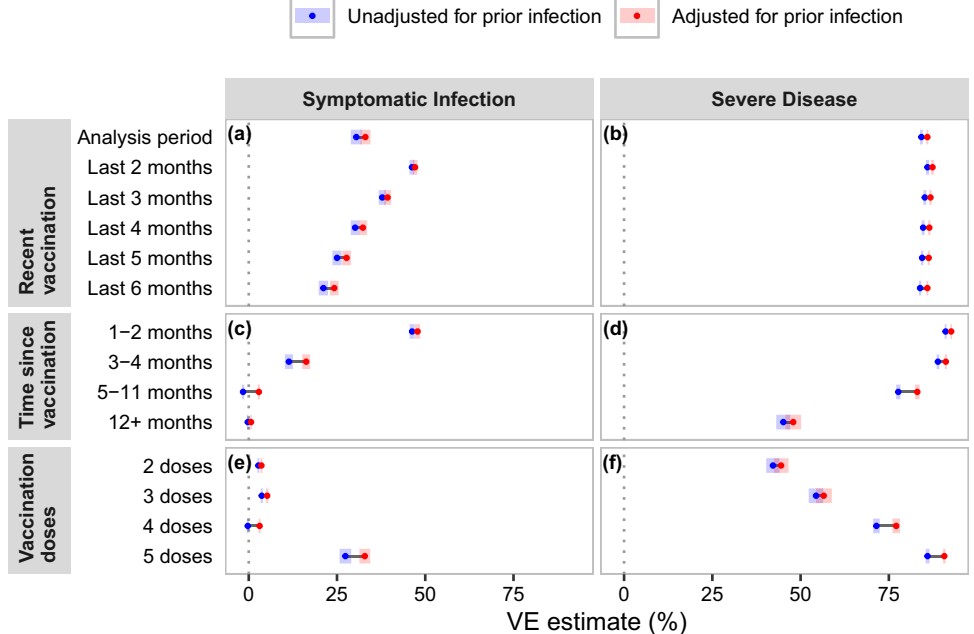

**Fig. 1 | Plot of estimated marginal means of VE against symptomatic infection and VE against severe disease for each exposure.** VE estimates are generated from a simple meta-regression of 768 simulation conditions, each summarized from 1000 simulations without controlling for simulation parameters. Estimates are presented as the marginal mean (as dots) +/− the 95% confidence interval (represented by bars) that are a product of the standard error and normal distribution quantiles. Panel identifiers are (**a**) recent vaccination exposures for VE against symptomatic infection; (**b**) recent vaccination exposures for VE against severe disease; (**c**) time since vaccination exposures for VE against symptomatic infection; (**d**) time since vaccination exposures for VE against severe disease; (**e**) vaccination dose exposures for VE against symptomatic infection; and (**f**) vaccination dose exposures for VE against severe disease.

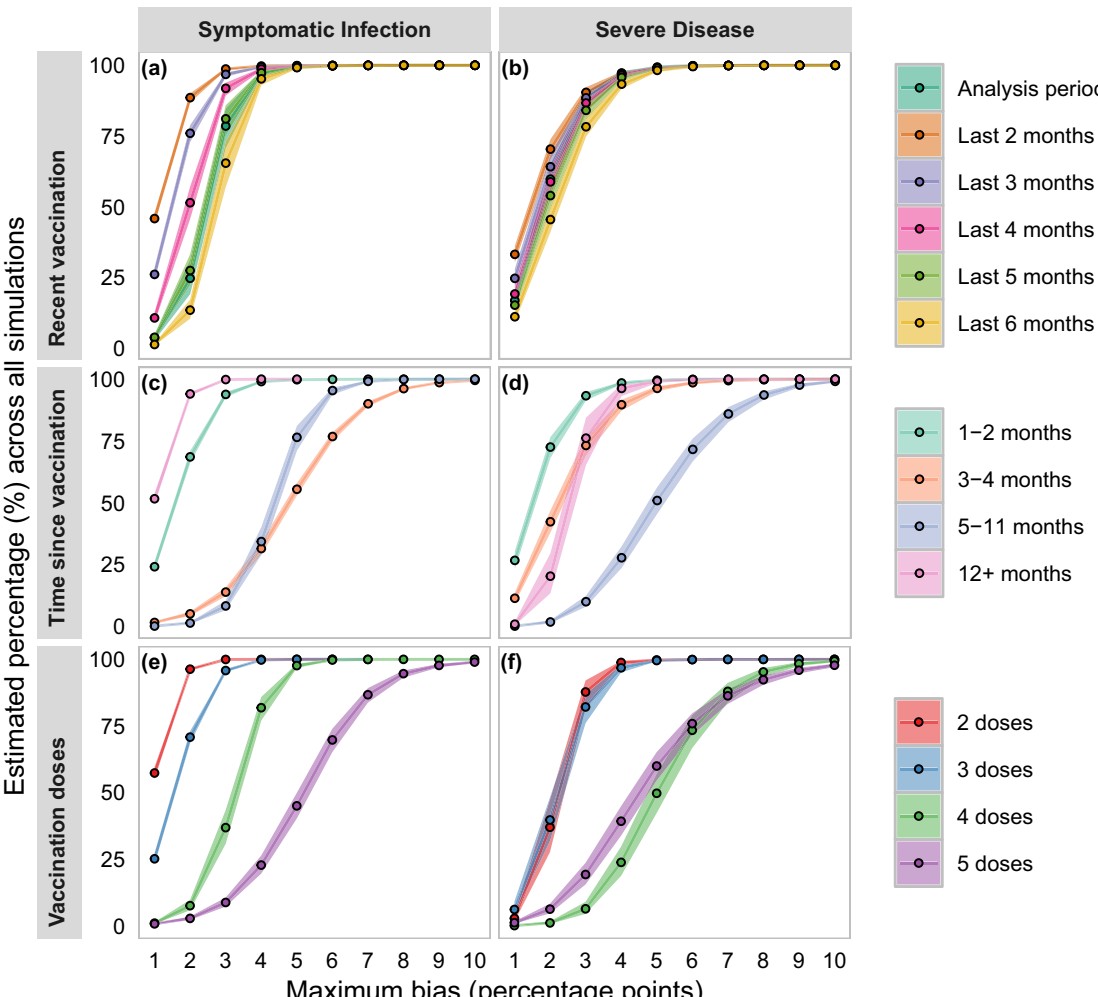

**Fig. 2 | Plot of estimated percentage of bias less than or equal to a percentage point threshold for VE against symptomatic infection and VE against severe disease for multiple exposures.** Bias was computed as the difference between VE calculated from the model that does not adjust for prior infection ("unadjusted") and the model adjusted for prior infection ("adjusted"). Bias estimates were generated from a meta-regression of aggregated results from 768 simulation conditions, each of which was summarized from 1000 simulations. Estimates are presented as the marginal mean estimate (as dots) +/− the 95% confidence interval (represented by bands connecting the maximum percentage point bias thresholds) that are a product of the standard error estimate and normal distribution quantiles. Panel identifiers are (**a**) recent vaccination exposures for VE against symptomatic infection; (**b**) recent vaccination exposures for VE against severe disease; (**c**) time since vaccination exposures for VE against symptomatic infection; (**d**) time since vaccination exposures for VE against severe disease; (**e**) vaccination dose exposures for VE against symptomatic infection; and (**f**) vaccination dose exposures for VE against severe disease.

21−32, respectively (Bias = −0.9 pp; CI: −1.4, −0.5), though the confidence interval overlapped with multiple other timing combinations and all combinations were over 99% likely to be biased no more than 6 pp.

In addition, the timing of the vaccination rollout in relation to the TND influenced VE against symptomatic infection estimates. For people vaccinated in the previous 3 months, VE against symptomatic infection from unadjusted models was nearly 20 pp lower when the vaccination rollout immediately preceded the TND (vaccination rollout in weeks 1−12 and TND in weeks 11−12: VE = 25.4%; CI: 24.6, 26.3; vaccination rollout in weeks 11−22 and TND in weeks 21−32: VE = 26.4%; CI: 25.6, 27.3) compared to when vaccination rollout and TND overlapped (weeks 1−12: VE = 46.0%; CI: 45.5, 47.0; weeks 11−22: VE = 45.1%; CI: 44.4, 45.7; weeks 21−32: VE = 46.2%; CI: 45.5, 47.0) (Fig. 3).

Other exposure definitions also attributed the largest differences in bias to the hybrid protection definition and the timing of the vaccination rollout and TND (Supplementary Fig. 4−16), though the waning of vaccine-induced protection also impacted bias and the likelihood of bias being below 6 or 8 pp for multiple exposures (Supplementary Figs. 4, 5, 8−11, 15).

The timing of the vaccination rollout and TND was the only factor which contributed to unadjusted VE against symptomatic infection being negative for people vaccinated in the previous 3 months (Fig. 3). Negative VE against symptomatic infection was most likely when the vaccination rollout happened after the TND, which is similar to performing a TND long after a vaccination campaign was completed (0.2% when vaccination rollout in weeks 11−22 and TND in weeks 1−12 and vaccination rollout in weeks 21−32 and TND in weeks 1−12) which was similar to exposures with a long time since vaccination, e.g., in people 12 or more months since vaccination which had at least 40% of VE estimates below zero (Supplementary Fig. 30).

## VE against severe disease
The median protection against severe disease at the end of the historical period ranged from 0.87 to 0.97 and the distribution of median protection was higher when hybrid protection was boosted by 30% compared to when VP was boosted by 30% of IP or IP was boosted by

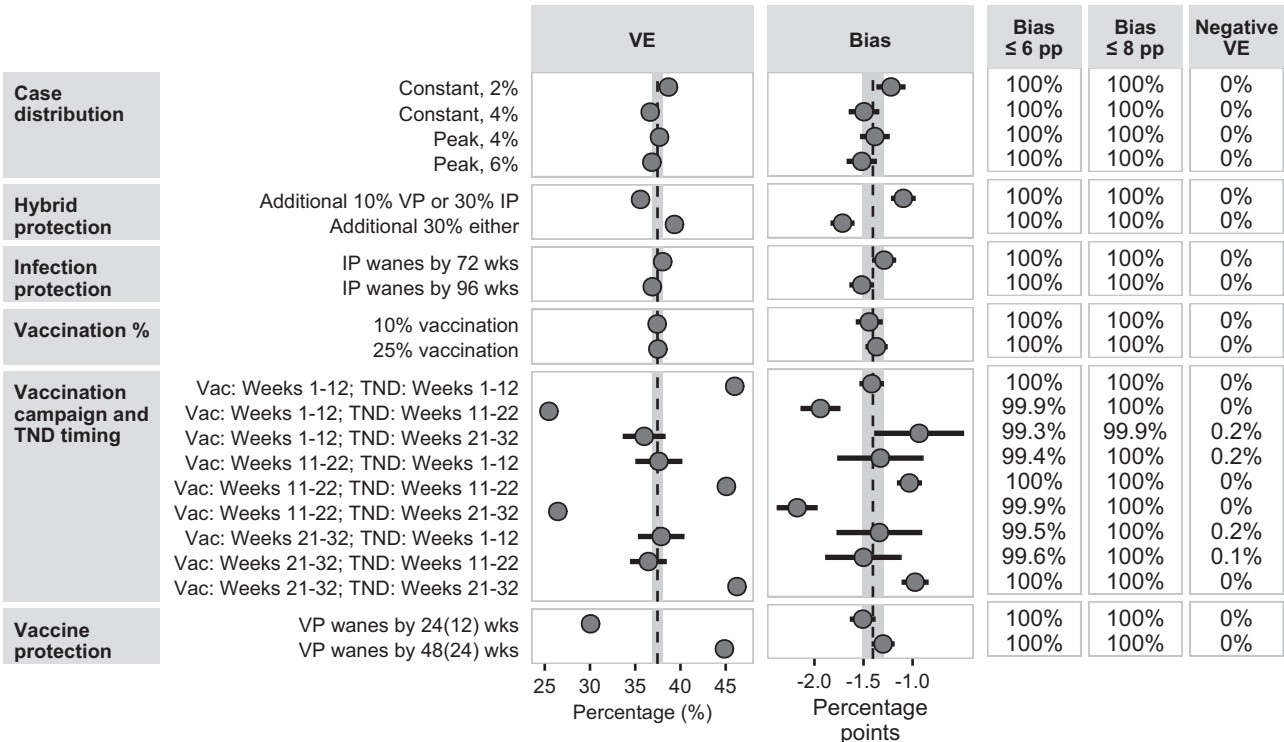

**Fig. 3 | Plot of estimated marginal means of unadjusted VE (not controlling for prior infection) against symptomatic infection, bias compared to adjusted VE (controlling for prior infection) against symptomatic infection, the percentage of simulations with a bias less than or equal to 6 percentage points (pp), 8 pp, and VE estimate less than zero (negative VE) for people vaccinated in the previous 3 months.** Estimates were generated from a meta-regression of aggregated results from 768 simulation conditions (each of which was summarized from 1000 simulations) after controlling for all other simulation parameters. Bias was computed as the difference between the unadjusted VE estimate compared to the VE estimate adjusted for prior infection. Estimates are presented as the marginal mean (as dots) +/− the 95% confidence interval (represented by solid lines) that are a product of the standard error and normal distribution quantiles. The bars may be narrower than the dot and not visible. The dashed line and shaded region in the VE column represent the overall mean and the 95% confidence interval, respectively, from Fig. 1. In the Bias column, the dashed line and shaded region represent the overall mean and the 95% confidence interval, respectively, from Supplementary Fig. 21.

10% of VP (Supplementary Table 1). The distribution of median protection against severe disease increased with increasing number of vaccinations (Supplementary Table 2) and was lower for those without a prior infection compared to any number of prior infections (Supplementary Table S3).

VE against severe disease (Fig. 1, panels b, d, f) in unadjusted models was highest for people 1−2 months since vaccination (VE = 91.1%; CI: 90.8, 91.3) and lowest for people 12 or more months since vaccination (VE = 42.2%; CI: 40.3, 44.1). For recent vaccination definitions, VE against severe disease had a small range from 87.4% (CI: 87.0, 87.7) for vaccination in the last 2 months to 85.9% (CI: 85.6, 86.2) for vaccination in the last 6 months.

Unadjusted models were at least 92% likely to underestimate VE against severe disease by at most 8 pp across all vaccination exposures (Fig. 2, panels b, d, f). The likelihood of bias of VE against severe disease being at or below 6 pp was above 98.5% for all recent vaccination exposures, 2 and 3 vaccination doses, and 1−2, 3−4, and 12+ months since vaccination. Mean bias of unadjusted models for VE against severe disease was no greater than 5.1 pp (Supplementary Fig. 3).

Overall bias for VE against severe disease for those vaccinated in the previous 3 months (Fig. 4) was −1.8 pp (CI: −2.0, −1.6). Bias for all parameter levels overlapped with those limits, except bias was less when a constant 4% case distribution was assumed (Bias = −2.0 pp; CI: −2.3, −1.7). Bias associated with all parameter levels was less than or equal to 6 pp in at least 99.3% of simulations. Other vaccination exposure definitions (Supplementary Fig. 17−29) also demonstrated differences in bias by variable levels, including by case definition (Supplementary Fig. 17, 19, 22, 25, 28, 29), hybrid protection (Supplementary Fig. 25−28), and vaccine-induced protection definition (Supplementary Fig. 17, 20, 21, 23−29).

## Discussion

These microsimulations suggest that, when many people have experienced at least one prior infection, failure to adjust for infection-induced protection does not dramatically change VE estimates from a TND. On the aggregate, across an array of exposure definitions, VE against symptomatic infection and VE against severe disease were underestimated by less than 8 percentage points in over 99% of simulations for most exposure definitions. Biases of between 6 to 8 percentage points in TNDs have been considered minimal enough to use for vaccine policy making[19−22], and, as has been argued previously, biases toward 0% should not restrict the utility of a VE estimate as a downward biased VE estimate may provide a lower bound[13].

Though, the aggregated results mask variability between parameter combinations. First, for simulation parameters, the bias of VE against symptomatic infection was impacted by the timing of the vaccination rollout and TND. The association between bias and timing varied by exposure but tended to be lowest when the vaccination rollout and TND were contemporaneous and largest when the vaccination rollout started three months prior to the TND. The increase in bias may be due to increased time since vaccination since vaccination was most likely early in the 12-week vaccination period, indicating vaccine-induced protection waned before the TND. Differential depletion of susceptibles[14,23] may also be a factor since vaccination is assumed to offer limited protection against infection and higher protection against severe disease.

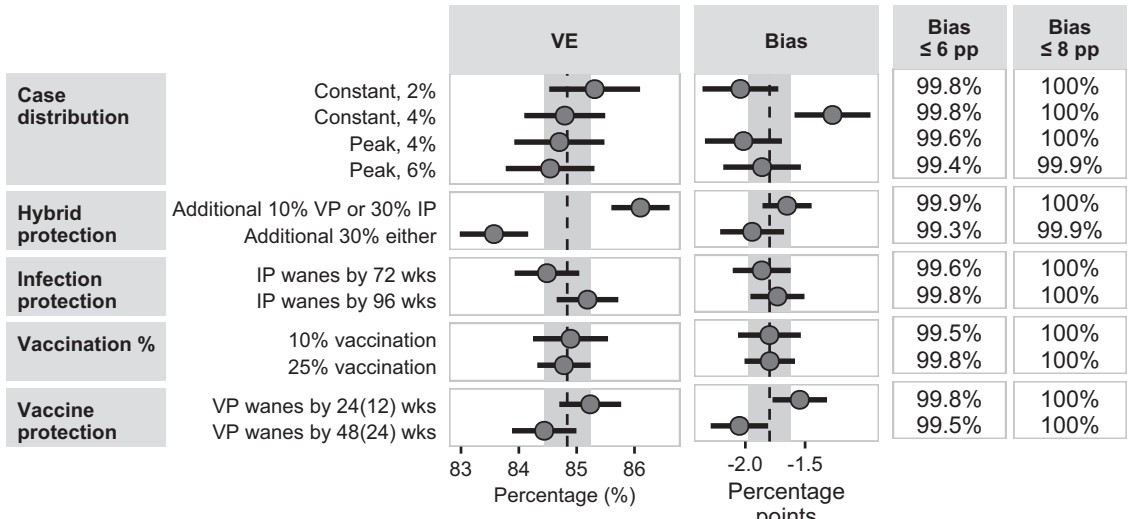

**Fig. 4 | Plot of estimated marginal means of unadjusted VE (not controlling for prior infection) against severe disease, bias compared to adjusted VE (controlling for prior infection), the percentage of simulations with a bias less than or equal to 6 percentage points (pp), or 8 pp for people vaccinated in the previous 3 months.** Estimates were generated from a meta-regression of aggregated results from 768 simulation conditions (each of which was summarized from 1000 simulations) after controlling for all other simulation parameters. Bias was computed as the difference between the unadjusted VE estimate compared to the VE estimate adjusted for prior infection. Estimates are presented as the marginal mean (as dots) +/− the 95% confidence interval (represented by solid lines) that are a product of the standard error and normal distribution quantiles. The dashed line and shaded region in the VE column represent the overall mean and the 95% confidence interval, respectively, from Fig. 1. In the Bias column, the dashed line and shaded region represent the overall mean and the 95% confidence interval, respectively, from Supplementary Fig. 3.

Second, bias for exposure and parameter combinations was as low as −13.2 pp (CI: −18.1, −8.8). In total, 150 exposure and parameter combinations possessed a bias of less −8 pp out of 10,752 total combinations (1.4%). One hundred of those were from VE against symptomatic disease simulations where the vaccination rollout occurred before the TND period. The most common exposures with a bias of less than −8 pp were 4- or 5-dose VE in 40 parameter combinations. These results suggest recognizing the entire context and all parameters is important to understanding the potential bias.

The timing of the vaccination rollout and TND also affected VE against symptomatic infection. VE against symptomatic infection for vaccination in the previous 3 months was ~46% with concurrent vaccination rollout and TND, 27% when rollout immediately preceded the TND, and 38% otherwise. These results suggest an impact for VE against COVID-19 symptomatic infection, potentially of 20 percentage points. Since VE against symptomatic infection wanes quickly, understanding the relative timing of the TND and vaccination rollout is critical for estimating VE for all exposures.

VE against symptomatic infection less than zero (negative VE) was more likely for exposure groups with more months since the last vaccination dose or fewer vaccination doses. Waning of vaccination-induced protection is a potential contributor to negative VE estimates[24,25]. Vaccinated individuals further from their last vaccination dose or with fewer doses have vaccination-induced protection that has completely or near-completely waned, which is likely driving the negative VE estimates in these exposures. This is especially true for symptomatic infection since waning may mean vaccinated individuals can be at a similar or greater risk of a mild outcome with SARS-CoV-2 infection compared to unvaccinated individuals during the TND since unvaccinated individuals are more likely to have a prior SARS-CoV-2 infection compared to vaccinated individuals[12], indicating that unvaccinated people are at greater likelihood of protection unaccounted for in unadjusted analyses compared to vaccinated people. As a comparison, VE against severe disease had no lower confidence limits below zero since VE against severe disease is greater than VE against symptomatic infection, and VE against severe disease wanes at a much slower rate than VE against symptomatic infection. Vaccine protection waning and existing infection-induced protection in unvaccinated participants suggest a higher outcome rate may be observed in vaccinated TND participants compared to unvaccinated TND participants, leading to a negative VE estimate. In addition, scenarios where a TND was performed three months after the vaccination rollout had the greatest likelihood of negative VE, further supporting vaccine-induced protection waning as a contributor to negative VE estimates. Bias also can contribute to negative VE[15], and we found a positive VE in adjusted analyses of < 6% could be underestimated in unadjusted analyses enough to bias an estimate below zero. Finally, random variation may also play a role and some exposures from individual parameter sets with a VE against symptomatic infection point estimate above 40% had a lower confidence intervals below zero. Therefore, exposure categories further out from the last vaccination possessed a high enough VE estimate to avoid the underestimation from unadjusted models resulting in a negative VE.

Our finding that VE estimates unadjusted for prior infection remain reliable and thus can be used to inform policy is especially important as prior infection is challenging to accurately measure. For example, adult VE studies from the US during SARS-CoV-2 Omicron variant circulation found ~15% of included patients with prior documented or self-reported laboratory-confirmed SARS-CoV-2 infection during a period when the vast majority of adults in the US had serological evidence of past infection[26,27]. A number of factors are likely to contribute to this, including asymptomatic or paucisymptomatic infection[28] that does not prompt testing, a lack of clinical testing despite symptomatic illness, receiving a prior positive test for SARS-CoV-2 in settings not captured in the surveillance network such as a different healthcare system and at-home testing[29,30], and imperfect accuracy of SARS-CoV-2 diagnostic assays. In addition, while a binary indication of prior infection may be available via serology in some study platforms, infection-induced protection is likely related to the number of prior infections, variant of prior infection(s), and time since prior infection, none of which are indicated via serology or fully captured by electronic health records or self-reporting.

The results also suggest that, when measuring VE for recent vaccination exposures, VE against severe disease is more stable than VE against symptomatic infection due to the slower waning of protection against severe disease. For all evaluated durations of the recent vaccination exposure, unadjusted VE against severe disease ranged from 83.9% to 85.9% whereas VE against symptomatic infection ranged from 21.2% to 46.2% indicating that the choice of recent exposure definition had less impact on VE against severe disease compared to VE against symptomatic infection.

The lack of a clear function of how infection-induced and vaccine-induced protection combine to become hybrid protection was one of the multiple limitations of these simulations. We utilized published meta-analyses that attempted to characterize the waning effectiveness of vaccines[31] and hybrid protection[32,33], but we required additional assumptions for our simulations. There is rich information on antibody titer trajectories[34,35] but challenges remain for determining the relationship between neutralization titers and protection[36]. We tried to create realistic simulations that were also succinct and understandable. As a result, we did not incorporate other known sources of bias, such as errors in vaccine registry linkage[37] or correlation between COVID-19 and influenza vaccination[20]. Another major consequence of creating realistic simulations was the true VE was dependent on the population in each simulation. Therefore, bias in these simulations was not based on a true, underlying parameter. We also did not vary the population size, which may affect the uncertainty of bias estimates.

In addition, although we found differences in the bias associated with vaccination doses, likely this was attributable to the timing of the last vaccination dose. Fewer vaccination doses were typically associated with a longer duration since the last vaccination dose. Therefore, in this simulation, people with fewer doses possessed less vaccine-induced protection and were more likely to have overall protection levels similar to unvaccinated people.

TNDs have been recommended as the most efficient and feasible method for assessing VE[38]. The effectiveness of vaccinations delivered is based not only on the vaccine formulations and the circulating pathogens but also on the characteristics of the population, including people's underlying immunity from past infections. Prior SARS-CoV-2 infections, including the number, variant, and timing of past infections, cannot be ascertained with certainty and are more common in unvaccinated compared to vaccinated individuals[12]. Although VE estimates unadjusted for prior infection are lower than adjusted estimates, the difference was in line with accepted underestimation of VE. Extra care should be taken when performing analyses by number of total vaccine doses as more recent doses have the potential for greater bias when not controlled for past infection and doses further in the past have greater potential to result in negative VE estimates. Ideally, researchers could adjust VE estimates from a TND for prior infection history if data are available, but unadjusted VE estimates from a TND remain useful.

## Methods

### Simulation methods

A thorough summary of the simulation methods and a full list of microsimulation parameter sets and results are included in the supplementary materials. We created populations of 100,000 people aged 18–49 years without protection against SARS-CoV-2 infection at the beginning of the COVID-19 pandemic (the week of January 19, 2020)[39]. Each week until the week of May 7, 2023, we updated each person's vaccine- and infection-induced protection against SARS-CoV-2 infection based on their most recent infection and vaccine dose since people could accumulate multiple infections and doses over time (Fig. 5).

Weekly infection probabilities were derived from aggregated case count data from 60 U.S. jurisdictions[39] divided by 2020 population estimates[40] (Supplementary Fig. 30). These proportions were increased by a multiplier (Supplementary Fig. 31) to account for underreporting of infections[41] and to reach ~95%–98% of the

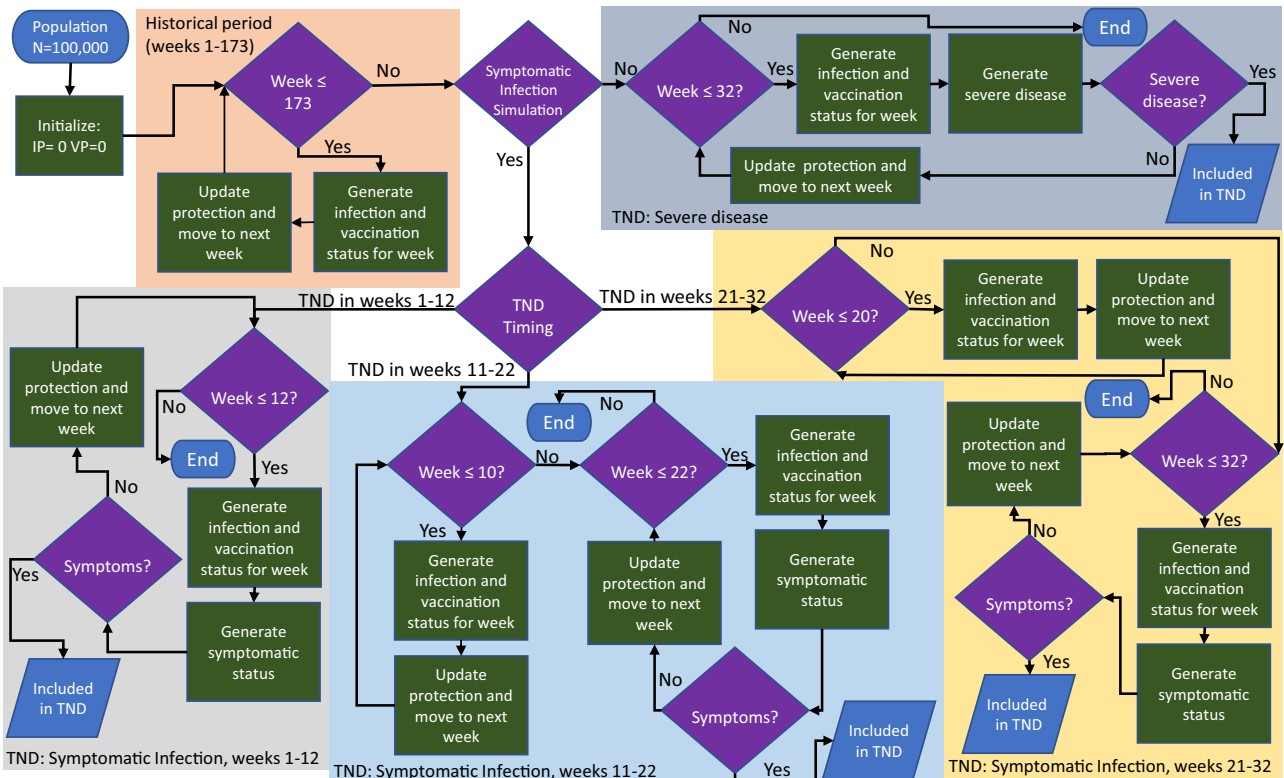

**Fig. 5 | Diagram of the simulation process.** IP=infection-induced protection, VP=vaccination-induced protection, TND=test-negative design.

population acquiring a prior infection by the end of the study period (Supplementary Fig. 32).

Individual, weekly probabilities of vaccination receipt utilized U.S. vaccination data, the number of prior doses and prior infection status. Data on vaccination distributions by vaccination dosage and week for U.S. people aged 18–49 years[42] (Supplementary Fig. 33) were fit to probability distributions (Supplementary Fig. 34). For naïve people, we set the probability of obtaining two vaccination doses at 0.70, the probability of a third dose conditional on having two doses at 0.30, and the probability of a fourth dose conditional on having a third dose at 0.10[42]. People with a prior SARS-CoV-2 infection have been less likely to initiate or subsequently receive an additional vaccination dose[43–48]. We assumed people with a prior infection were less likely to receive an additional dose with an odds ratio of 0.525[43–48].

A person's protection was based on the most recent week of vaccination and infection, based on product-limit survival curve estimates from the third dose efficacy trial of the BNT162b2 vaccine (Pfizer–BioNTech), where the placebo and vaccine curves started to diverge after approximately one week[49], and usage of a one-week lag in recent VE studies[5,50–53]. Waning curves were based on trajectories in published literature[24,31,32,54–56]. A week after vaccination, we assumed 90% vaccine-induced protection against infection (VP) prior to Omicron predominance and 70% protection thereafter. VP waned linearly to zero (1) at 48 weeks post-vaccination prior to Omicron predominance and 24 weeks thereafter or (2) at 24 weeks post-vaccination prior to Omicron predominance and 12 weeks thereafter (Supplementary Fig. 35) with variability by person (Supplementary Fig. 36). A week after infection, infection-induced protection (IP) had 90% protection against infection that waned to zero at 96 or 72 weeks (Supplementary Fig. 37) again with variability by person (Supplementary Fig. 38).

Hybrid immunity or protection (HP) definitions were taken from meta-analyses of protective effectiveness[32,33]: (1) the greater of VP or IP was boosted by 30% of the other (Supplementary Fig. 39); or (2) VP was boosted by 30% of IP or IP was boosted by 10% of VP, whichever was greater (Supplementary Fig. 40). Both HP definitions were truncated at 99%. In these simulations, we considered 8 different protection calculations since we simulated each combination of the two VP, two IP, and two HP definitions.

Infections were generated from a person's weekly protection with the function

$$\Pr\left(I_{j,k}\right) = \Pr(c_k) * \left(1 - \psi_{j,k-1}\right), \qquad (1)$$

where the probability of infection for each person ($j$) and week ($k$), $\Pr\left(I_{j,k}\right)$, depended on $\Pr(c_k)$, the case probability in week $k$ and person $j$'s protection calculated from the previous week ($\psi_{j,k-1}$). An infection for person $j$ in week $k$ was generated from a Bernoulli distribution with probability $\Pr\left(I_{j,k}\right)$.

A total of 200 populations were generated for each of the 8 protection definition combinations. An example of protection trajectories is included in the supplementary materials (Supplementary Fig. 41).

The analytic period consisted of a hypothetical 32-week period beginning immediately after the historical period. Infections, vaccination doses, and protection were generated similarly to the historical period. Parameters were the 8 protection definition combinations, case distribution, vaccination rollout timing, total vaccination coverage, TND timing, and type of outcome (symptomatic infection or severe disease).

Four infection distributions were utilized during the analytic period (Supplementary Fig. 42): weekly 2%; weekly 4%; weekly 2% increase to a peak of 4% at weeks 16 and 17 before returning to 2%; and

weekly 2% increasing to a peak of 6% at weeks 16 and 17 before returning to 2%. The vaccination rollout happened in weeks 1–12 (before the case peak), weeks 11–22 (during the case peak), or weeks 21–32 (after the case peak) and followed a lognormal distribution with a mean of 1.5 and a standard deviation of 0.5 (Supplementary Fig. 43). Other weeks had a vaccination probability of 0.005. End-of-season vaccination coverage in the analytic period alone was 10% or 25% (Supplementary Fig. 44) to provide parameters above and below the estimated 14% coverage in people aged 18–49 years in the 2023–2024 season[57]. The TND for symptomatic infections was implemented in weeks 1–12, weeks 11–22, or weeks 21–32. Since we implemented all possible combinations of vaccination rollout and TND timing, some scenarios involve assessing VE via the TND before the vaccination rollout. These scenarios approximated the situation where VE is assessed long after vaccination has been given. All 32 weeks were used for the TND for severe disease.

COVID-19 symptoms were expected in 80% of infected people (Supplementary Fig. 45) and were present only in the week of infection. An uninfected person in week $k$ was expected to have COVID-like symptoms with a probability of 0.20 divided by the number of weeks in the TND (Supplementary Fig. 46). For estimating VE against symptomatic infection, all symptomatic people were included in the TND. Diagnostic testing was assumed to have perfect specificity, but sensitivity was 90% during the week of infection and declined thereafter[58] (Supplementary Table 4).

For estimating VE against severe disease, VP was 90% the week after vaccination and waned to zero after 48 months[24,31,56,59]. IP against severe disease started at 95% protection the week after infection and waned to zero after 96 months[32] (Supplementary Fig. 47). For people with a SARS-CoV-2 infection, the probability of severe disease was

$$\Pr\left(S_{j,k} | I_{j,k} = 1\right) = \frac{\left(1 - \psi^s_{j,k-1}\right)}{\left(1 - \psi_{j,k-1}\right)}, \qquad (2)$$

where $S_{j,k}$ is a severe disease event for person $j$ in week $k$, and $\psi^s_{j,k-1}$ is the protection against severe disease for person $j$ in week $k$. All people with severe disease were included in the TND with perfect detection.

A total of 1000 simulations were run for each parameter set. Each of the 200 populations was utilized five times in each parameter set.

## Statistical methods

Exposures analyzed were vaccination at any time during the analytic period, vaccination in the previous 2 months, vaccination in the previous 3 months, vaccination in the previous 4 months, vaccination in the previous 5 months, vaccination in the previous 6 months, the time since vaccination (unvaccinated as the reference group, 0–2 months, 3–4 months, 5-11 months, and 12 or more months), and the number of doses (unvaccinated as the reference group, 2-dose, 3-dose, 4-dose, or 5-dose) where someone with 5 doses received all available vaccination doses, a person with 4 doses missed one of the available vaccination doses, and so on.

Two logistic regression models were fit to each exposure definition. The first model included only the exposure variable (henceforth, the unadjusted model), whereas the second model added categorical time since the last infection (categories were monthly from 1 to 11 months and 12 or more months) and the number of prior infections as a continuous variable (the adjusted model). Odds ratios (OR) from logistic regressions were converted to VE in percentage points by the formula

$$VE = (1 - OR) * 100. \qquad (3)$$

Our primary measure is the difference between the VE estimate from the unadjusted model and the VE estimate from the adjusted

model, which we refer to as bias. Bias is defined not in the traditional sense as the deviation from truth, but as the percentage point difference in VE from the unadjusted model and VE from the adjusted model. Bias less than zero indicated VE was underestimated without accounting for prior infection. A small percentage of simulations resulted in small sample sizes and unstable estimates. Details on bias definition and handling of unstable estimates are in the supplementary methods.

Results were aggregated by parameter set and exposure and plotted by exposure with ridgeline plots (Supplementary Fig. 2). Simple, random effects meta-regression was used to estimate the expected VE and bias and, for infection outcomes, the percentage of simulations with a negative VE estimate. Separate meta-regressions were run for the unadjusted and adjusted VE estimates. Multivariable meta-regression models were run with simulation parameters to determine the mean VE, bias, and negative VE associated with each parameter level and the 95% confidence intervals. Sensitivity analyses were performed for VE against symptomatic infection by removing scenarios where the TND was implemented before the vaccination rollout.

All simulations were performed in R version 4.0.4, and analyses in R version 4.2.4.

### Reporting summary
Further information on research design is available in the Nature Portfolio Reporting Summary linked to this article.

## Data availability
No empirical data were used in the analyses of this manuscript. The simulated data generated for this study can be created using scripts provided as a compressed file in the Supplementary Information.

## Code availability
Programming code used in the project is available as a compressed file (Supplementary Code 1). Please refer to the readme PDF in the compressed file for more information.

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

## Acknowledgements

The findings and conclusions in this report are those of the authors and do not necessarily represent the views of the CDC. This study was supported by the Centers for Disease Control and Prevention. B.F.'s time was supported by the Centers for Disease Control and Prevention contract number 75D30120C07765 to Kaiser Foundation Hospitals. We thank clearing officials at CDC and anonymous reviewers for improving this manuscript.

## Author contributions

R.L.G. conceived the idea. R.E.W. supervised the project, implemented the simulation study, analyzed the results, and prepared the manuscript. R.E.W., B.F., M.N., M.W.T., R.L.G., and J.M.F. contributed to the study design, discussed the results, read the manuscript and supplement, and provided critical feedback.

## Competing interests

The authors declare that they do not have any commercial or other associations that might pose a conflict of interest.
