## [Transparent Peer Review file · Nature Communications]

Bias and negative values of COVID-19 vaccine effectiveness estimates from a test-negative design without controlling for prior SARS-CoV-2 infection

Corresponding Author: Dr Ryan Wiegand

Version 0:

Reviewer comments:

Reviewer #1

(Remarks to the Author)

Summary:

This paper sought to estimate the often-unmeasured bias of prior infection on COVID-19 vaccine effectiveness utilizing simulation methods. The authors explore a variety of situations in which VE may be impacted by prior infection and found that the bias in estimation of prior infection in VE calculation may be small. This is an important and relevant topic as prior infection is an important but difficult to measure consideration when estimating vaccine effectiveness.

Major Comments:

General

Confidence intervals are not meaningful here as they are simulated by population size, making them arbitrary. Instead it would be useful to see a more formal analysis of the % of simulations with a bias of x,y,z percentage points.

Methods

Line 100 – could the authors explain the rationale behind utilizing 1 week instead of 2 weeks? Many studies have utilized a 2 week threshold before calculating immunity.

Line 100 – 90% protection against infection seems wildly too high, and this is a very important parameter. Could the authors adjust this value and/or provide justification for such a high value?

Line 105 – Are these values among already vaccinated people? Figure S8 shows “weeks since vaccination”, should this say “weeks since infection”? If not, would this graph instead represent hybrid immunity?

Line 120 – what is the exact timeframe of the historical period? This does not appear to be defined in either the main manuscript or in the supplement. Figure 1 is a little confusing as to exactly when this period ends.

Line 129 – could the authors provide a justification for using 4th dose vaccination coverage as the total vaccination coverage? Is this total vaccinations occurring within the analytic period, or does this include vaccinations during the historical period?

Results

Line 231 – this sounds backwards: please clarify: “Negative VE against symptomatic infection was most likely when the vaccination rollout happened after the TND, which is similar to performing a TND long after a vaccination campaign was completed”.

Line 231 – Were simulations in which VE was calculated ahead of vaccine rollout included in overall results? Could the authors provide a justification as to why this was done if so?

Discussion

I would be interested in seeing the authors' comments regarding to what extent these results apply to vaccines against other imperfectly-immunizing infections – and specifically those that are endemic where there will be a lot of prior infection in the prior to vaccination

To what extent do these results apply to other study designs? TND is most common for effectiveness studies, but traditional case-control and cohort studies are also used.

Minor Comments:

Line 48 – remove extra period

Line 213 – should this say weeks 11-22? Analyses following show an 11 week TND vs this 1 week TND. If it should show as a 1 week TND, could the authors explain why this period was shorter than other analyses?

Line 271 – not a complete thought

(Remarks on code availability)

Reviewer #2

(Remarks to the Author)

In this paper, the authors perform a lengthy simulation study to examine the potential for bias in COVID-19 vaccine studies due to missing information on prior infection. If available, data on documented prior infections are known to be unreliable as prior infections are often not documented. This presents a challenge in estimating vaccine effectiveness as unvaccinated individuals (e.g., individuals without the most recent dose under study) may have a different immune history than vaccinated individuals, and they may be more likely to have infection-induced immunity, leading to under-estimated VE. Through their simulations, the authors estimate that this bias is unlikely to exceed 8%, which reassures policy-makers that observational studies, despite their limitations, are not overly affected by this bias in the range of settings explored in this simulation study. In my view, this is the biggest contribution of the article. Other noteworthy results are that negative VE estimates are most likely to occur for studies estimating symptomatic infection using an exposure group with more months since the last vaccination dose.

The methodology is reasonable. One can choose from a wide range of parameters in the simulations, and of course there are combinations of parameters not explored, but the authors were intentional about their choices and justified these. And the simulation study is already quite large as is, so I won't suggest more.

Comments:

- First, I'll note that the paper is not an easy read. It is dense and dry. As someone interested in this area, I am a target reader, but even I had some trouble extracting the major conclusions.
- Second, the authors stuck to realistic parameters based on this setting, but, given their work on this paper, do they have intuition about future settings either in COVID or in other disease areas where this bias could even be larger? What if some of the variability in willingness to get vaccinated were more exaggerated? In what extremes might this change?
- The authors compare different ways of coding exposure to vaccination. Can they make any stronger statements about the preferred option for future studies?

Minor comments:

- Figure 2: The big difference for the 5th dose is because it is the "recent" dose in the study. I guess the comment is making sure people understand that there isn't something special about 5, it just reflects that many individuals in the simulation are on their 5th dose.
- Figure 4: What are the vertical dotted lines? Not specified in figure caption or note.
- Line 332 "indicating that bias of VE against severe disease was less likely to be influenced by the exposure duration." Meaning unclear.
- Legend for Supplement Figure S12 hard to understand.

(Remarks on code availability)

Reviewer #3

(Remarks to the Author)

(Remarks on code availability)

Version 1:

Reviewer comments:

Reviewer #1

(Remarks to the Author)

The authors have done a thorough and comprehensive job responding to my review. All of my concerns have been satisfactorily addressed.

(Remarks on code availability)

NA

Reviewer #2

(Remarks to the Author)

I'm satisfied with the authors' revisions. Still a dense read, but the edits have moved it in the right direction.

(Remarks on code availability)

October 11, 2024

Ryan E. Wiegand
Centers for Disease Control and Prevention
MS H24-5
1600 Clifton Road NE
Atlanta, GA 30329 USA

REVIEWER COMMENTS

Reviewer #1 (Remarks to the Author):

Major Comments:

General

Confidence intervals are not meaningful here as they are simulated by population size, making them arbitrary. Instead it would be useful to see a more formal analysis of the % of simulations with a bias of x,y,z percentage points.

Response: We agree with the underlying sentiment of this comment that multiple parameters may affect the confidence interval width, e.g., the population size, probability of infection and hospitalization, and the likelihood of COVID-19-related symptoms. Since the focus of this manuscript is on prior infection and the effect that derived infection-induced protection may have on VE, we focused on parameters we felt would directly influence protection. Thus, we did not vary population size and other parameters even those all of those could impact the number of people sampled in the test negative design (TND), the standard errors, and confidence interval width.

To address this comment, we moved Figure 3 to the supplement (now Figure S21) and replaced it with a figure containing the estimated likelihood that bias will be less than a threshold. We implemented thresholds of 1, 2, ..., 10 percentage points (pp) and Figure 3 has dots for each threshold and lines connecting them with 95% confidence intervals from the meta-regressions.

In addition, Figures 4 and 5 have added columns for the percentage of unadjusted VE estimates with bias ≤ 6 pp and bias ≤ 8 pp, per this suggestion. In these figures, we also converted the negative VE forest plot panel to a column of percentages so all results could fit in the figure.

Though, for Figures 4 and 5, we would like to propose a compromise for the forest plots of bias estimates. Our results section and discussion depend on this panel to show which estimates differed from the overall bias. This allows us to highlight which factor levels increase or decrease bias. Thus, we would appreciate it if the forest plots could remain in this panel for this purpose.

In the text, we focused on the thresholds of 6 pp and 8 pp, but the results from each of the bias pp thresholds are included in all relevant supplementary plots and the supplementary output.

The paragraphs on lines 206-219 and 261-272 have been revamped to focus on the percentage with bias less than or equal to a threshold. This comment also prompted changes or additions to the text in lines 230, 240-241, 275-276, 285-286, and 364-365.

Methods

Line 100 – could the authors explain the rationale behind utilizing 1 week instead of 2 weeks? Many studies have utilized a 2 week threshold before calculating immunity.

Response: This is a valid question and we don't think there is a definitive answer here. Our utilization of 1 week after vaccination stems from the clinical trial results of BNT162b2 (<https://www.nejm.org/doi/full/10.1056/NEJMoa2200674>) where the Kaplan-Meier curves for the placebo and vaccine groups looked to diverge around 7 days post dose (Figure 3, copied below).

[REDACTED]

This differs from the curves for BNT162b2 after the second dose (<https://www.nejm.org/doi/full/10.1056/nejmoa2034577>)

[REDACTED]

and for mRNA-1273 after two doses (<https://www.nejm.org/doi/full/10.1056/nejmoa2035389>)

[REDACTED]

which both had much longer time for the curves to separate. Some recent vaccine effectiveness analyses continue to use 14- or 21-day lags, with some of those decisions made to be consistent with prior studies (e.g., <https://doi.org/10.1016/j.lana.2022.100198>), but a 7-day lag has been used in multiple recent studies of COVID-19 VE. We added our justification for this choice to lines 99-102 with references of recent papers using a 7-day lag.

Line 100 – 90% protection against infection seems wildly too high, and this is a very important parameter. Could the authors adjust this value and/or provide justification for such a high value?

Response: Citations for our protection values are provided in line 102. Specifically, Tartof et al ([https://doi.org/10.1016/S0140-6736\(21\)02183-8](https://doi.org/10.1016/S0140-6736(21)02183-8), Figure 2) found approximately 90% VE against infection and hospital admission at <1 month, Britton et al (<https://doi.org/10.1001/jama.2022.2068>, text) found an OR of 0.10 which translates to a VE of 90% for adults 14-60 days since second dose, and Andrews et al (<https://doi.org/10.1056/NEJMoa2119451>, Table 3) found approximately 90% VE against symptomatic disease caused by the Delta variant for 2- and 3-doses. Feikin et al’s meta regressions ([https://doi.org/10.1016/S0140-6736\(22\)00152-0](https://doi.org/10.1016/S0140-6736(22)00152-0), Figure 2) are a little confusing to interpret but it looks like 90% VE at time zero seems like a reasonable assumption.

Though, please note that the 90% VE is protection *pre-Omicron*. In line 104, we note that we drop the protection to 70% which aligns with the available information when we implemented the simulations (e.g., Ciesla et al. <https://doi.org/10.1093/ofid/ofad187>).

Line 105 – Are these values among already vaccinated people? Figure S8 shows “weeks since vaccination”, should this say “weeks since infection”? If not, would this graph instead represent hybrid immunity?

Response: Thank you for noticing this. The label on Figure S8’s x-axis should say “weeks since infection” and has been corrected.

Line 120 – what is the exact timeframe of the historical period? This does not appear to be defined in either the main manuscript or in the supplement. Figure 1 is a little confusing as to exactly when this period ends.

Response: The timeframe for the historical period is defined in lines 81-84, specifically the MMWR week of January 19, 2020, through the MMWR week of May 7, 2023 (the end of the public health emergency).

Line 129 – could the authors provide a justification for using 4th dose vaccination coverage as the total vaccination coverage? Is this total vaccinations occurring within the analytic period, or does this include vaccinations during the historical period?

Response: The total vaccination coverage is the probability of anyone in the analytic period receiving a dose, i.e., an end-of-season coverage. This does not include vaccinations in the historical period, with the caveat that those without a prior infection would still be more likely to obtain a vaccination.

We chose 10% since early trends of the 2023-2024 vaccination coverage suggested end-of-season coverage would be much lower than prior years. Coverage for 2023-2024 vaccinations in the U.S. for those aged 18-49 years was approximately 14% (https://data.cdc.gov/Vaccinations/Cumulative-Percentage-of-Adults-18-Years-and-Older/hm35-qkiu/about_data), which suggests our 10% estimate underestimated by 4 percentage points. We've changed "Total vaccination coverage" to "End-of-season vaccination coverage" in line 130, changed text in lines 131-133, and changed the reference to reflect the current season coverage.

Results

Line 231 – this sounds backwards: please clarify: "Negative VE against symptomatic infection was most likely when the vaccination rollout happened after the TND, which is similar to performing a TND long after a vaccination campaign was completed".

Response: This is stated correctly. Negative VE has been a topic of importance for us and others that participate in WHO's COVID-19 VE methods forum. Thus, one of the stated goals of these simulations was to address this question in the context of these simulations. We wanted to include a condition where the TND was conducted after the effectiveness of vaccines had completely waned. Implementing the TND immediately before the vaccination rollout would have the same effect and be more efficient compared to running the simulation for multiple months. This is why the language seems backwards. This scenario was described in the simulation methods section and we have updated lines 135-137 to make this clearer.

If you have any suggestions for improved language, we would greatly appreciate it.

Line 231 – Were simulations in which VE was calculated ahead of vaccine rollout included in overall results? Could the authors provide a justification as to why this was done if so?

Response: Yes, all simulations were included these estimates. There is no rationale to including all simulations.

Nevertheless, given this and your prior comment, it would appear as though you have some discomfort with including all simulations into the estimates. Thus, we have performed sensitivity analyses for VE against symptomatic infection where only simulations where the TND is during or after the vaccination rollout are included (Figure S21). This figure uses the same results as the new Figure 3, specifically the estimated likelihood that bias will be less than thresholds of 1, 2, ..., 10 percentage points (pp) with dots for each threshold and lines connecting them with 95% confidence intervals from the meta-regressions. Though, this figure is structured differently where each exposure has its own panel and the panel rows are the exposure categories. Each plot has two lines, one line with all simulations (which can also be found in Figure 3) and a second line for the same results but without simulations where the TND happened after the vaccination rollout.

The lines for all simulations and only simulations with TND before vaccination rollout are indecipherable once the threshold reaches 6 pp (except for the 5-doses exposure) suggesting there is no difference at maximal, policy-relevant thresholds. There are some larger differences at lower thresholds, such as at 3 or less pp for the last 5 months, last 6 months, 4 doses, and 5 doses. Also, when there are differences

between the two estimates, sometimes the all simulations estimates is consistently lower (e.g., 4-doses) and sometimes the TND before vaccination rollout estimate is consistently lower (e.g., 5-doses). Thus, there was no estimate that was consistently more conservative.

We have noted the existence of these sensitivity analyses in the statistical methods subsection (lines 184-185) and added a sentence in the manuscript (lines 216-218) briefly describing the results.

Discussion

I would be interested in seeing the authors' comments regarding to what extent these results apply to vaccines against other imperfectly-immunizing infections – and specifically those that are endemic where there will be a lot of prior infection in the prior to vaccination

To what extent do these results apply to other study designs? TND is most common for effectiveness studies, but traditional case-control and cohort studies are also used.

Response: We were careful to not overinterpret and overextend these results to other diseases and methods when creating these simulations and composing this manuscript. Our reasons for doing that stem from factors which limit the portability of these results to other diseases. For instance, seasonality differs between diseases, e.g., the influenza season tends to be shorter (often 3-4 months) than SARS-CoV-2 seasonality thus far. Seasonal influenza also has three types or subtypes that are currently circulating (H1N1, H3N2, and B/Victoria) which don't always circulate in predictable ways. For instance, during the 2023-2024 season, B/Victoria had much higher circulation relative to other B viruses over the past 3 seasons. Therefore, if immune response varies to each type/subtype, the protection conferred by vaccination may differ across populations and impact VE against influenza-associated outcomes differently than VE against COVID-19-associated outcomes.

Plus, the magnitude of VE waning may differ between influenza and COVID-19, with some suggestion that VE against symptomatic influenza infection may wane slower (<https://doi.org/10.1093/cid/ciw816>) than COVID-19 VE against symptomatic infection (e.g., <https://doi.org/10.1093/ofid/ofad187>). Therefore, questions around the impact of waning are less pertinent for influenza compared to COVID-19 since COVID-19 (so far) has tended to have year-round transmission and peaks twice a year. Second, the impact of a COVID-19 prior infection on COVID-19 protection may differ from the impact of prior infection on other diseases. We made assumptions in these simulations that were highly dependent on COVID-19 protection that may not apply to other diseases.

Finally, we felt uncomfortable extending these results to other designs. A cohort study may have more information on past infections since participants should have information collected over a longer period of time. Therefore, our assumption has been that a cohort study should have some information on past infection. That would indicate the impact of missing prior infection information should be less from a cohort study. Though, missing prior infection information may also indicate missing follow up time too, which and we are unsure the impact of that missingness on a Cox model or other methods used to analyze cohort data.

A matched case-control design is an interesting idea and may be something we could consider in the future. Ultimately, this manuscript is quite long and having to incorporate another methodological approach might turn this into a 6,000 word manuscript, which is probably unwise (see reviewer 2's "dense and dry" comment).

Minor Comments:

Line 48 – remove extra period

Response: Removed. Thank you!

Line 213 – should this say weeks 11-22? Analyses following show an 11 week TND vs this 1 week TND. If it should show as a 1 week TND, could the authors explain why this period was shorter than other analyses?

Response: Thank you! That should say “22” instead of “12”. That has been changed (now line 226 in the manuscript).

Line 271 – not a complete thought

Response: We’re guessing that this comment comes from the use of “mask”. Mask is used as a verb to indicate that the aggregated results hide or cover up the variability of the individual parameter sets. We’ve modified this statement slightly to try to provide clarity (now line 290).

Reviewer #2 (Remarks to the Author):

The methodology is reasonable. One can choose from a wide range of parameters in the simulations, and of course there are combinations of parameters not explored, but the authors were intentional about their choices and justified these. And the simulation study is already quite large as is, so I won’t suggest more.

Response: Thank you!

Comments:

- *First, I’ll note that the paper is not an easy read. It is dense and dry. As someone interested in this area, I am a target reader, but even I had some trouble extracting the major conclusions.*

Response: We endeavored to make this paper relevant to a wide audience. A pure VE paper or a pure methods paper are easier to write (and read) because the audience is smaller. Since we are trying to reach a broad readership, our plan was to shift as many of the gory details to the supplement; though, to do the paper justice, we still have a methods section that is ~1400 words. Based on some comments from reviewer 1, we shortened the results which may help a little bit.

Nevertheless, based on your summary, it seems like you were able to pull out the main ideas. Sorry you had to dig for them, though.

• *Second, the authors stuck to realistic parameters based on this setting, but, given their work on this paper, do they have intuition about future settings either in COVID or in other disease areas where this bias could even be larger? What if some of the variability in willingness to get vaccinated were more exaggerated? In what extremes might this change?*

Response: Our response to reviewer #1's comment on p.6 that starts, "I would be interested in seeing the authors' comments...", summarizes our feelings about not extrapolating these results to other disease areas. In brief, there are too many differences between COVID-19 and other diseases for us to feel comfortable extrapolating these results.

Our feeling is that there are ways in which bias could increase, especially anything that may modify how cases and controls are ascertained. For instance, one area that worries us are rapid diagnostic tests (RDTs) since the scenario where a person with a symptomatic infection tests positive via an RDT and then doesn't present to a medical facility seems likely to us. If usage of RDTs differs between cases and controls or vaccinated and unvaccinated, that could add to the bias we have seen in these simulations. This is mentioned in Shi et al. (<https://doi.org/10.1093/aje/kwac203>) and is a goal of ours to look at this.

The willingness to get vaccinated is a good point since this could cause further selection bias. Our feeling is that parameter did not have as much of an effect as it could have since so many people had a prior infection by the analytic period. Other behaviors that drive the cases to look different from the controls in aggregate is a concern.

Beyond the study design decisions (e.g., the outcome, timing of the TND and vaccination rollout, the exposure definition), the largest bias for individual parameter sets we explored was when people had the greatest chance to have residual protection. Therefore, it would appear that bias might increase in situations with slower waning for vaccine- and infection-induced protection. Though, if waning is more gradual, the vaccine starts to approach the profile of a sterilizing vaccine instead of one with waning, which would necessitate different analyses

• *The authors compare different ways of coding exposure to vaccination. Can they make any stronger statements about the preferred option for future studies?*

Response: The choice of vaccination exposure categories should be driven by the vaccine policy questions. These questions have differed over time which is partially why has led to different ways of categorizations of vaccination exposure in analyses, and makes a preferred, single recommendation somewhat challenging. Though, as COVID-19 research and policy progresses, policy may coalesce around specific questions which may also lead to more standardization in how vaccination is coded. When we started these simulations, there were still many different exposure definitions being used, hence, the wide array of exposure definitions included in the manuscript.

In more recent analyses, we have included both an analysis of vaccinated or not vaccinated along with a time since last vaccination analysis. The reasons for each are because there is interest in seeing an "overall" estimate of VE, but also wish to see estimates of the vaccination waning. Thus, we'd probably be most inclined to vote for using the "Analytic period" and "Time since vaccination" categories since those most closely align with the current questions. Though, the questions may change and our choice may have to change as a result.

Minor comments:

- *Figure 2: The big difference for the 5th dose is because it is the “recent” dose in the study. I guess the comment is making sure people understand that there isn’t something special about 5, it just reflects that many individuals in the simulation are on their 5th dose.*

Response: The difference for the people with 5 doses compared to all other dose categories is that we can guarantee that they have had a recent dose since the only way someone could have 5 doses is if she/he/they has had every dose offered. We edited lines 158-162 by moving the time since vaccination exposures before the dose exposures and then adding more details so it is clear what these categories represent.

- *Figure 4: What are the vertical dotted lines? Not specified in figure caption or note.*

Response: Thanks for catching this. The explanation was in the text, but not in the figure notes, which is probably the better place for that description. We’ve cut the text in lines 181-184 and have added that information (with some minor changes) to lines 608-610 and lines 618-620.

- *Line 332 “indicating that bias of VE against severe disease was less likely to be influenced by the exposure duration.” Meaning unclear.*

Response: We edited the text on lines 352-354 to try to make this statement clearer.

- *Legend for Supplement Figure S12 hard to understand.*

Response: Thanks for your thorough review! I (the lead author) had put in abbreviations as a placeholder and I am disappointed I forgot to change this. The legends in Figure S12 have been updated to include labels that are more descriptive and in line with the text.

Reviewer #3 (Remarks to the Author):

Response: Welcome! Hopefully it was a good experience.